# Integrin αv and Vitronectin Prime Macrophage-Related Inflammation and Contribute the Development of Dry Eye Disease

**DOI:** 10.3390/ijms22168410

**Published:** 2021-08-05

**Authors:** Tsung-Chuan Ho, Shu-I Yeh, Show-Li Chen, Yeou-Ping Tsao

**Affiliations:** 1Department of Medical Research, Mackay Memorial Hospital, New Taipei City 25160, Taiwan; hoct1295@yahoo.com.tw; 2Department of Medicine, Mackay Medical College, New Taipei City 25245, Taiwan; shuiyeh@gmail.com; 3Department of Ophthalmology, Mackay Memorial Hospital, Taipei 10449, Taiwan; 4Department and Graduate Institute of Microbiology, College of Medicine, National Taiwan University, Taipei 10617, Taiwan; showlic@ntu.edu.tw

**Keywords:** dry eye disease, inflammation, macrophage, αv integrin, vitronectin, c(RGDfK)

## Abstract

Cell signaling mediated by the αv integrin plays a pivotal role in macrophage activation in various inflammatory processes, but its involvement in the pathogenesis of dry eye disease (DED) remains unclear. In a murine model of DED, we found increased αv integrin expression in ocular surface macrophages. The αv integrins inhibitor c(RGDfK) ameliorated the corneal damage caused by DED, suggesting a pathogenic role for αv integrin. Because tear hyperosmolarity induces ocular inflammation in DED, a hyperosmolar culture of murine bone marrow-derived macrophages (BMDMs) is used to reproduce inflammation in vitro. However, the expression of proinflammatory cytokine mRNA was minimal, even though αv integrin was induced. In searching for components that are involved in αv integrin-mediated inflammation but that are missing from the culture model, we showed that the levels of vitronectin (VTN), a binding ligand of αv integrins, were increased in the tear fluid and conjunctival stroma of DED animals. The addition of VTN prominently enhanced hyperosmolarity-induced inflammation in BMDMs. Mechanistically, we showed that VTN/αv integrins mediated NF-κB activation to induce inflammatory gene expression in the BMDMs. Our findings indicate that interaction the of VTN with αv integrins is a crucial step in the inflammatory process in DED and suggests a novel therapeutic target.

## 1. Introduction

Dry eye disease (DED) is a multifactorial disorder of the ocular surface characterized by symptoms of discomfort, visual disturbance, decreased tear quality, and chronic inflammation [1]. Reduced tear secretion and/or the increased evaporation of tears leads to tear film instability and tear hyperosmolarity that are associated with ocular surface inflammation and corneal epithelial injury in DED [2].

Macrophages represent a heterogeneous cell population (M1, M2, and others) [3,4]. High salt (NaCl) culture medium used to reproduce tear hyperosmolarity has been used to facilitate the generation of pro-inflammatory M1 macrophages [4,5]. Using a murine model with DED induced by a desiccating environment and scopolamine to maximally reduce tear production, the distribution of pro-inflammatory M1 macrophages on the ocular surface has been investigated, and M1 macrophages were found to predominantly reside in the conjunctiva [6]. Notably, the subconjunctival injection of clodronate to locally deplete the resident conjunctival macrophages significantly reduces ocular surface damage and proinflammatory cytokine expression in dry eyes [7,8]. A mouse model of desiccating stress-induced dry eyes is shown to recruit monocytes into the conjunctiva and to induce them into activated macrophages [7,8]. The activated macrophages have upregulated expression of the genes that are associated with antigen presentation, cytokines/chemokines, M1 macrophages, and the NLRP3 inflammasome pathway [8]. These activities further enhance the inflammatory status of the ocular surface. The crucial involvement of macrophages in ocular surface inflammation is based on the depletion of macrophages. Subconjunctival injection of clodronate to locally deplete the resident conjunctival macrophages significantly reduces the generation of autoreactive CD4^+^ T cells, ocular surface damage, and proinflammatory cytokine expression in dry eyes [9,10]. Therefore, macrophages maybe a novel drug target for patients with DED.

Integrins are transmembrane receptors and present asαβheterodimers. They execute extensive functions by interacting with a range of extracellular ligands, including vitronectin (VTN) [11,12]. VTN contains an Arg-Gly-Asp (RGD) sequence that intervenes by binding to five distinctαv integrins (αvβ1, αvβ3, αvβ5, αvβ6 and αvβ8) and the platelet receptor αIIbβ3 to promote cell attachment and migration [9,10]. Notably, αv integrin has been studied in various disease models. For example, αvβ3 integrin has been shown to be highly expressed in activated macrophages in atherosclerotic plaques, but levels of expression on quiescent macrophages in the normal aorta are minimal [13]. In addition, the depletion of the αv integrin in hepatic stellate cells (HSCs) protects mice from hepatic fibrosis, whereas the depletion of theβ subunit alone has no such effect [14]. Consistently, microglial cells with aβ3/β5-double knockout can respond normally to VTN-mediated cell attachment and inflammatory responses because theβ1 and β8 can bind theαv subunit, suggesting a compensatory role of β subunits in the αv integrins-related effects [15]. The expression of macrophage αv integrin in DED remains unclear. Therefore, exploring the expression of αv integrin in the macrophages may be helpful to address the underlying mechanism of DED pathogenesis.

In this study, we used a murine DED model to characterize the expression αv integrin in the macrophages. We found that the expression of the αv integrin was dysregulated in the macrophages residing in the conjunctiva. Meanwhile, there was more VTN in the tear fluid and the conjunctival stroma of DED mice than in mice housed in a non-stressed (NS) environment. The importance of αv integrins/VTN signaling in the pathogenesis of experimental DED was evaluated using c(RGDfK), an inhibitor of αv integrins. In addition, we used murine BMDMs as an in vitro model, and our findings showed that the αv integrins were induced in BMDMs when the cells were cultured in the NaCl hyperosmotic medium. The induction of αv integrin in the macrophages enhanced the VTN-stimulated inflammatory responses. Currently, treatment for DED has several unmet medical needs, including the need for more effective and specific anti-inflammatory therapy [16]. Our findings are the first to suggest an underlying mechanism by targetingαv integrins to resolve macrophage-related inflammation in DED.

## 2. Results

### 2.1. High Salt Upregulates the Gene Expression of αv Integrins in Macrophages

High salt (NaCl) has been used to activate the mouse BMDMs, with a shift towards apro-inflammatory M1 phenotype, characterized by the upregulation of the inducible nitric oxide synthase (iNOS) and interleukin (IL)-1β and the downregulation of a well-known M2-associated marker, arginase 1 (Arg-1) [4,5]. As shown in Figure 1a, the qPCR results supported previous studies; the BMDMs were cultured in hyperosmotic medium with the addition of various doses of NaClfor 24 h. Treatment with 10~40 mM NaCl increased the mRNA expression of *iNOS* (3.6~5.1-fold), compared to untreated BMDMs. In contrast, BMDMs treated with 20 and 40 mM NaCl had 2.1-fold and 2.3-fold lower levels of *Arg-1* than untreated BMDMs. However, NaCl treatment did not significantly affect the baseline mRNA expression of *IL-10*. In addition, the 40 mM NaCl treatment did not induce cell death, as monitored by a trypan blue exclusion test (our unpublished data, not shown).

The relationship between high salt and the expression of αv integrins in the macrophages remains unclear. BMDMs were treated with 10~40 mM NaCl for 3 h and 24 h. Subsequently, qPCR analysis revealed that the levels ofαv, β1, and β5 mRNA were 2.4~3.6, 2.3~3.6, and 1.9~4.0-fold greater than in untreated cells after NaCl treatment for 3 h, and this increased further to 5.8~9.4, 4.2~7.1, and 3.8~9.8-fold following 24 h induction (Figure 1b). The levels of integrin β3 and β8 mRNA were not altered during the 3 h treatment but were significantly upregulated to 1.7~3.7-fold and 1.7~2.9-fold compared to the untreated control at 24 h. However, integrin β6 expression was not induced. Immunofluorescence staining showed increased αv integrin levels in BMDMs after treatment with NaCl for 24 h (Figure 1c), and this was confirmed by Western blotting (2.9- ± 0.3 fold; Figure 1d). We determined then whether NaCl-mediated hyperosmolarity could increase αv integrin expression in human THP-1 macrophages. Exposure of the THP-1 macrophages to 10~40 mM NaCl for 3 h induced a 1.5~2.5-fold increase of αv integrin, compared to the untreated control, and this increased to 4.2~5.9-fold after 24 h. In addition, after NaCl treatment for 24 h the levels of αv integrin protein increased by~3.5-fold, compared to untreated cells (Figure 1f).

Taken together, αv integrin mRNA and protein expression in cultured macrophages are upregulated by hyperosmolarity induced by NaCl. In addition, theαv, β1, and β5 integrins display an early response (3 h) to stimulation with NaCl, suggesting that the increased hyperosmolarity has a direct effect through a transcriptional pathway to regulate the expression of the αv, β1, and β5 integrin genes.

### 2.2. Expression of the αv Integrin in Macrophages Is Induced in Dry Eyes

After observingαv integrin induction in cultured macrophages, we were interested in the potential of the dry eye environment to induce αv integrin expression in the macrophages. C57BL/6 mice were induced to generate DED as described in “Methods”. A previous study with a similar animal model showed that M1 macrophages are mainly found in the conjunctiva rather than in the cornea after DED induction for 10 days [6]. As shown in Figure 2a, immunofluorescence staining with iNOS and F4/80 showed that most of the F4/80-positive macrophages in the conjunctiva displayed an iNOS/M1-positivephenotype, similar to what has been previously reported. In contrast, almost all of the macrophages were weakly positive or negative for iNOS in the non-stressed (NS) control. Furthermore, immunofluorescence analysis revealed that the number ofαv integrin-positive macrophages was higher in the dry eye, approximately 2.3-fold more than the NS control (Figure 2b,c). Meanwhile, qPCR analysis of *iNOS* and *αv integrin* mRNA expression in the conjunctiva of the dry eye revealed amounts that were 2.7 ± 0.6-fold and 3.4 ± 0.5-fold higher than the NS group (Figure 2d). The results indicate a specific induction of the αv integrin in the conjunctival macrophages by experimental DED induction.

### 2.3. VTN Is a Proinflammatory Stimulator of BMDMs

BMDMs responding to NaCl (20 mM or higher) stimulation have been found to have increased expression of proinflammatory cytokines [4,5]. Our qPCR assay confirmed these findings (~1.5-fold increased, *p* < 0.05; Figure 3a). In addition to NaCl stimulation, VTN (100~500 ng/mL) treatment for 3 h was also able to induce the gene expression of *TNF-α*, *IL-1β*, and *IL-6*up to 1.5~3-fold, compared to untreated BMDMs (*^a^ p* < 0.05; Figure 3b–d), suggesting that VTN is a proinflammatory stimulator of the macrophages. Importantly, the VTN (100~500 ng/mL)-induced *TNF-α*, *IL-1β*, and *IL-6* expression was further elevated to 2.3~6.2, 3.5~5.7, and 2.4~4.9-fold when the BMDMs were treated with 20 mM NaCl for 24 h. These results imply that the high salt-induced M1 macrophages are more susceptible to VTN-induced proinflammatory responses than the resting macrophages.

### 2.4. Levels of VTN in the Tear and Conjunctival StromaareElevated in DED Mice

The interplay of VTN with αv integrins promotes BMDM-mediated proinflammatory gene expression. We investigated the expression and distribution of VTN in dry eyes. Using an ELISA assay, we found that slightly raised VTN protein concentrations in tear fluid after DED induction for 1 week, and these further increased by nearly 2-fold after DED induction for 2 and 3 weeks, compared with the NS mice (230 ± 22 and 229 ± 18 versus 114 ± 7 ng/mL, Figure 4a). VTN was found at high concentrations in blood serum [17] but was not elevated in the sera of DED mice. Immuno-histochemical staining of VTN revealed strong immuno-reactivity in the conjunctival stroma of the dry eye. In contrast, staining of the NS control was inconspicuous (Figure 4b). In addition, qPCR and Western blot analysis of the *VTN* gene and protein expression in conjunctiva isolated from DED mice were 2.3-fold and 2.9-fold greater than the NS mice, respectively (Figure 4c,d), suggesting that the local biosynthesis of VTN in the conjunctiva is stimulated in dry eyes.

### 2.5. c(RGDfK) Benefitsthe Recovery of Experimental Dry Eye

To extend our understanding of the potential pathogenic role of αv integrins in DED, their inhibitor c(RGDfK) was used to treat the experimental DED. As depicted in Figure 5a,b, after DED induction for 8 days, the corneal epithelial barrier displayed obvious injury, which was detected by fluorescein staining. Mice with fluorescein staining scores of 2 were judged to have moderate/severe DED and were randomized to receive either c(RGDfK) or vehicle treatment by subconjunctival injection on days 8 and 10. Prior to treatment (day 8), the scores in the vehicle and c(RGDfK) groups showed no statistically significant difference (score = 2.5 ± 0.1). Treatment with c(RGDfK) for 5 days (Day 12) led to a significant decrease in the fluorescein staining score compared to the vehicle group (1.9 ± 0.2 versus 2.9 ± 0.1; *p* = 0.03). This indicates that c(RGDfK) can ameliorate pre-existing ocular surface damage in the dry eye. Goblet cells are predominantly found in the superficial epithelium of the conjunctival fornix and are responsible for mucous tear production. Periodic acid–Schiff (PAS) staining of goblet cells in NS mice displayed a continuous homogeneous pattern at the conjunctival fornix (Figure 5c). However, PAS staining of the vehicle-treated DED mice showed a marked decrease of goblet cells, compared to the c(RGDfK) and NS groups (number/0.1 mm^2^: 3.5 versus 5 and 5.5), indicating that c(RGDfK) is able to prevent goblet cell loss during dry eye development. In addition, prior to treatment (day 8), tear production was significantly reduced in DED mice, compared to NS mice, as measured by a cotton thread test (4.1 ± 0.2 mm versus 6.1 ± 0.2 mm; Figure 5d). c(RGDfK) treatment prevented the decrease of the tear fluid to a level close to, but less than, normal levels at day 12 (5.1 ± 0.3 mm). Collectively, our findings reveal that c(RGDfK) has a therapeutic effect upon experimental DED. The results also imply that the αv integrin-positive macrophages are a target of c(RGDfK).

### 2.6. VTN Interacts with αv Integrins to Induce NF-κB-Mediated Proinflammatory Cytokine GeneExpression in NaCl-Treated Macrophages

The transcription factor NF-kB is required for the maximal transcription of the *TNF-α*, *IL-1β*, and *IL-6*genes [18]. In the canonical NF-κB signaling pathway, NF-κB p65 subunit phosphorylation at Ser-536 (p-p65) is a critical event for the activation of NF-κB in the macrophages [19]. After observing the proinflammatory cytokine genes prominently induced in the VTN/NaCl-treated BMDMs, we investigated whether VTN can induce p65 phosphorylation (p-p65) in the NaCl-treated BMDMs. Western blot analysis revealed that VTN treatment for 20~60 min resulted in a 2.6~3.2-fold greater amount of p-p65 than the in untreated control (Figure 6a). Moreover, c(RGDfK) significantly blocked VTN-induced p-p65 to near baseline levels. qPCR analysis also revealed that VTN-induced *TNF-α*, *IL-1β*, and *IL-6* gene expression (set as 100%) decreased by 50.8%, 65.5%, and 59.2%, respectively, when cells were pretreated with 10 µM c(RGDfK) (Figure 6b). In cells pretreated with 20 µM c(RGDfK), these further decreased to 20.2%, 31.3%, and 23.1%, respectively. In addition, VTN-induced p-p65 was considerably suppressed by the αv integrin blocking antibody and SN50 (an inhibitory peptide of NF-κB) compared to cells pretreated with IgG control (2.1-fold and 1.8-fold; Figure 6c). The αv integrin blocking antibody and SN50 also effectively suppressed the VTN-induced *TNF-α*, *IL-1β*, and *IL-6* gene expression by 21~25% and 22~39%, compared to cells pretreated with the IgG control (Figure 6d). Collectively, the results indicate that NF-κB activation is critical for proinflammatory gene expression induced by VTN/αv integrins.

### 2.7. c(RGDfK) Suppresses NF-κB Activation in Macrophages and Proinflammatory Cytokine Expression in the Dry Eye

We investigated whether the numbers of p-p65 positive macrophages increased after DED induction. Immunofluorescence staining showed that the numbers of p-p65 positive conjunctival macrophages clearly increased in the dry eye, compared to the NS group (47 ± 4.6% versus 8 ± 2.0%; Figure 7a,b). The nuclear accumulation of p-p65 suggests that NF-κB-mediated proinflammatory gene transcription is activated in the conjunctival macrophages. Notably, c(RGDfK) treatment showed slight staining of p-p65 in the nuclei, and the p-p65 positive macrophage population was reduced to 15 ± 3.6%.

It has been reported that TNF-α andIL-1 blockers can ameliorate the severity of experimental DED [20,21], suggesting that diminished TNF-α andIL-1βbenefit DED therapy. qPCR analysis revealed that the mRNA levels of *TNF-α*, *IL-1β*, and *IL-6* in dry eyes were significantly up-regulated by 3.0, 2.6, and 3.3-fold, respectively, compared to the NS mice (Figure 7c–e). Treatment with c(RGDfK) apparently repressed the mRNA expression of *IL-1β*, *TNF-α*, and *IL-6* by factors of 1.6, 1.8, and 1.7-fold, respectively, compared to the vehicle-treated group. The inhibitory effect of c(RGDfK) on p-p65 and proinflammatory cytokine expression strongly implies that αv integrins and NF-κB signaling are involved in macrophage-associated inflammation in vivo.

## 3. Discussion

Macrophages are important cellular elements in chronic inflammation that drive DED progression, but the details of how macrophages shift to a proinflammatory phenotype are not completely understood. In this study, the dysregulation ofαv integrin was found to be a crucial event in macrophage-mediated inflammation, including in the activation of NF-κB and in the induction of proinflammatory cytokine gene expression. The pathological significance of αv integrins in the development of DED may be highlighted by increased VTN protein expression in dry eyes and in the therapeutic effect of c(RGDfK). Our findings suggest that the positive association betweenαv integrins and VTN may represent a molecular mechanism by which macrophages respond to hyperosmotic stress and contribute to DED pathogenesis.

To date there is little information about the expression and function of integrins in the macrophages during DED pathogenesis. We show increases of αv integrin expression in the BMDMs and the THP-1 macrophages following NaCl-induced hyperosmotic stimulation. The induction of αv integrins in cells seems to be associated with certain pathogenic stresses. For example, the macrophages located in the tonsils and spleen do not expressαv and β3 integrins, whereas macrophages present in atherosclerotic lesions display a high level of αvβ3 integrin expression [13,22]. The hypoxia-inducible factor (HIF) has been linked to the induction ofαvβ3 integrin expression in trophoblast stem cells and melanoma cells [23]. The key stress-associated transcription factors that are involved in the transcriptional regulation of αv integrin in macrophages for osmotic adaptation remain largely unknown. This aspect waits further study.

c(RGDfK) binds with high affinity to theαvβ3 integrin and with moderate affinity to other αv integrins [24]. In addition to theαv integrins, α5β1, α8β1, and αIIbβ1 belong to the RGD-binding family of integrins and are capable of binding to c(RGDfK) [9,10,22]. Therefore, the antagonistic effect of c(RGDfK) on individual RGD-binding integrins is relative rather than absolute. However, our in vitro data showed thatαv integrins display a pivotal role in VTN-induced proinflammatory cytokine expression in BMDMs because these inductions are reduced by ~80% following treatment with an αv integrin blocking antibody. Therefore, it seems reasonable that the αv integrins are the main target of c(RGDfK) in this in vitro study.

The effect of c(RGDfK) on DED animals is rather complex although we note that conjunctival macrophages display higher levels of αv integrin than NS mice. It may be expected that the substantial therapeutic effect of c(RGDfK) on DED may be, in part, due to suppression of the inflammatory effects of several RGD-binding integrins in various ocular surface cells. For example, the α5β1 integrin plays an important role in the inflammatory infiltration of neutrophils [25,26]. Corneal epithelial cells (CECs) have been shown to express the αvβ1 integrin and are a source of inflammatory cytokines in dry eyes [27,28]. Previous works also indicate that inflammatory Th17 cells express high amounts of the αvβ3 integrin, which is critical for sustaining a Th17 inflammatory phenotype, such as IL-17 production and IL-17-induced corneal epithelial barrier disruption in dry eyes [29,30]. Therefore, neutrophils, CECs, and Th17 potentially may be therapeutic targets of c(RGDfK). In addition, drug discovery research in the anti-integrin therapy includes the lymphocyte function-associated antigen 1(LFA-1; integrin αLβ2; a T cell surface receptor). In this regard, Lifitegrast blocks the association of LFA-1 with intercellular adhesion molecule 1 (ICAM-1), leading to the alleviation T cell-mediated inflammatory responses in DED [31]. Inflammatory cytokines in the dry eye are derived from multiple sources in addition to the macrophages. The exact role of the VTN/αv integrin-dependent signaling in DED pathogenesis remains to be clarified.

NF-κB plays a key role in regulating the transcription of macrophage inflammatory response-related genes [19,32]. Our findings also revealed that NF-κB activation is associated with the proinflammatory cytokine expression induced in macrophages by VTN/αv integrins according to the effects of the NF-κB inhibitor SN50. Similarly, inhibiting αv integrins using the antagonist S247 or αv/β3 siRNAs suppresses the shear stress-induced NF-κB activation in the endothelial cells [33]. Previous works have highlighted the essential role of the PI3 kinase/Akt pathway in the proinflammatory responses mediated byαv integrins in the macrophages [34]. FAK and Pyk2 kinases have been associated with integrin signaling in vascular diseases and NF-κB activation [35]. The detailed signaling cascade that is involved in NF-κB activation induced by VTN/αv integrins remains to be clarified.

Soluble VTN is predominantly produced in the liver by the hepatocytes and is found at high concentrations in blood serum [17]. VTN is also expressed in various tissues as an extracellular matrix component. VTN has been found to be induced in the capillaries to facilitate macrophage infiltration in the ischemic hindlimb [36]. In the human eye, VTN is synthesized locally in the retinal pigment epithelium–choroidal complex and is involved in the pathogenesis of age-related macular degeneration (AMD) [37]. However, little is known about the role of VTN in the development of DED. Our data revealed that the VTN protein was not restricted to the dry eye but could also be detected in the ocular tissues of NS mice, including the tear fluid and conjunctival stroma. The reasons for higher VTN levels in the dry eye are not clear from the present study. It may be that a higher vascular permeability induced by desiccating stress enabled the diffusion of VTN from the plasma. It is also possible that VTN is secreted by cells in the conjunctival stroma, as supported by our qPCR analysis. Whether *VTN* gene expression is induced in the endothelial cells of the conjunctiva under hyperosmotic environment awaits further investigation.

## 4. Materials and Methods

### 4.1. Materials

Recombinant Human Vitronectin (140-09) and macrophage colony-stimulating factor (M-CSF) were purchased from PeproTech (Rocky Hill, NJ, USA). Hoechst 33258 dye and all chemicals were from Sigma-Aldrich (St. Louis, MO, USA). Anti-integrin alpha V (ab179475) and anti-F4/80 antibody (ab6640) were obtained from Abcam (Cambridge, MA, USA). Anti-vitronectin antibody (GTX61399) was purchased from GeneTex (Taipei, Taiwan, ROC). Phospho-NF-κB p65 (Ser536) (#3033) and NF-κB p65antibody (#8242) were purchased from Cell Signaling Technology (Danvers, MA, USA). FITC-donkey anti-rabbit IgG and FITC-donkey anti-mouse IgG were purchased from BioLegend (San Diego, CA, USA). SN50 (a cell-permeable NF-κB inhibitory peptide) and c(RGDfK) were purchased from Selleckchem (Houston, TX, USA).

### 4.2. Cell Culture and NaCl Treatment

These culture methods for the mouse BMDMs and the THP-1 macrophages from the THP-1 monocytic cell line have been described in detail previously [38]. For the NaCl treatment of the BMDMs, the loosened BMDMs were removed with a cell scraper and were separated into single cells using a treatment with 2 mL trypsin (0.25% and 0.01% EDTA) for 5 min at 37 °C and were then collected by centrifugation (800× *g* for 5 min). A total of 2 × 10^5^ cells were transferred to a well of a 6-well plate and were cultured with 10% FBS medium supplemented with M-CSF (10 ng/mL) for 24 h. Subsequently, the BMDMs were incubated with 1% FBS medium with M-CSF for 8 h, and the isomolar medium was then replaced by hyperosmotic media with the addition of 10~40 mM NaCl for a further 3 h and 24 h, as previously described [5].

### 4.3. Induction of DED

C57BL/6 mice (female; 8-weeks-old) were housed in an animal room under temperature control (24–25 °C) and a 12:12 h light-dark cycle. Standard laboratory chow and tap water were available ad libitum. Experimental procedures were approved by the Mackay Memorial Hospital Review Board (New Taipei City, Taiwan, ROC) (project code: MMH-A-S-107-41, date: 1 January 2019–31 December 2021) and were performed in compliance with national animal welfare regulations (Council of Agriculture, ROC).DED was induced by housing the mice in a controlled environment chamber (CEC) and administering scopolamine for maximal ocular surface dryness as previously reported [6,39]. In brief, CEC conditions were controlled to provide a relative humidity of <25%, airflow of 10 L/min, and temperature of 24–25 °C for 24 h a day. Mice were exposed to the CEC for 12 days, and scopolamine hydrobromide (0.375 mg dissolved in 0.15 mL PBS per mouse; Escopan, Taiwan) was injected subcutaneously into the mice two times per day at day 3 and day 5. Untreated age- and sex-matched mice served as a control (NS mice group).

### 4.4. Clinical Examination of DED

To evaluate the effect of desiccating stress on ocular surface disruption, fluorescein (100 mg/mL; Alcon Laboratories, Inc., Fort Worth, TX, USA) was instilled via a micropipette into the inferior-lateral conjunctival sac (0.6 µL of 0.5% sodium fluorescein dissolved in 4.4 µL of PBS per mouse). After fluorescein staining for 15 s, mouse eyes were washed once with PBS and were then examined using a slit lamp microscope under cobalt blue light. Tear production was measured with phenol red-impregnated cotton threads (Zone-Quick; Oasis, Glendora, CA, USA). The threads were held with jeweler forceps and were placed in the lateral canthus for 60 s. The tear production was expressed in the millimeters of thread that were made wet by the tears and turned red. The validity of this test in mice was performed as previously described [39].

### 4.5. PAS Staining of Goblet Cells

After the animals were euthanized, the eyes were surgically excised, fixed in 4% paraformaldehyde (PFA), were paraffin embedded, and were cut into 5-µm sections. The sections were stained with PAS (Sigma-Aldrich) reagent for measuring goblet cells in the superior and inferior conjunctiva. Images were captured using a Nikon Eclipse 80i microscope (Nikon Corporation, Tokyo, Japan) equipped with a Leica DC 500 camera (Leica Microsystems, Wetzlar, Germany). PAS-positive goblet cells in the conjunctiva were measured in five sections from each eye.

### 4.6. Quantitative Real-Time PCR

The total RNA extraction, cDNA synthesis, and qPCR were performed as described previously [38]. Primers used in the experiment are listed in Table 1.

### 4.7. Western Blot Analysis

Cell lysis, SDS–PAGE, and antibodies used for immunoblotting were as described in our previous study [38]. Antibodies used in this study were those against VTN, integrinαv, Phospho-NF-κB p65, and NF-κB p65 (1:1000 dilution). The band intensity in the immunoblots was evaluated with a Model GS-700 imaging densitometer and analyzed using Labworks4.0 software. (Bio-Rad Laboratories, Hercules, CA, USA).

### 4.8. Immunohistochemistry

PFA-fixed, paraffin-embedded DED specimens were deparaffinized in xylene and rehydrated in a graded series of ethanol concentrations. The slides were blocked with 10% goat serum and 5% bovine serum albumin (BSA) in PBS containing 0.5% TritonX-100 (PBST) for 60 min and were then incubated with the primary antibody against VTN (1:100 dilution) at 37 °C for 3 h. The slides were subsequently incubated with peroxidase-labeled goat immunoglobulin (1:500 dilution) for 20 min and were then incubated with chromogen substrate (3,3′-diaminobenzidine) for 2 min before counterstaining with hematoxylin.

### 4.9. Immunofluorescence

Deparaffinized tissue sections or 4% PFA-fixed BMDMs were blocked with 10% goat serum and 5% BSA in PBST for 20 min. Staining was performed using primary antibodies against F4/80 (1:100 dilution) at 37 °C for 3 h. For the staining of the αv integrin (1:100 dilution) or the phospho-NF-κB p65 (1:100 dilution), sections were stained at 4 °C overnight. The slides were subsequently incubated with the appropriate fluorescent-labeled secondary antibodies (1:500 dilution) at 37 °C for 1 h and were then counterstained with Hoechst 33258 for 6 min. The slide was viewed with an epifluorescence microscope (Zeiss Axioplan 2 imaging; Zeiss, Oberkochen, Germany) equipped with a charge-coupled device camera (Zeiss AxioCamHRm, Zeiss, Germany), and quantification was performed using Axiovert software (Zeiss AxioVision Release 4.8.2, Zeiss, Germany).

### 4.10. Mouse VTN ELISA

The concentration of VTN in the tear fluid was determined by ELISA. A balance salt solution (BSS; Alcone) was instilled onto the inferior fornix of a mouse at weekly intervals. The tear-washing fluids of the two eyes (each 5 µL; referred to here as tear specimens) in the same mouse were pooled together and were stored at -80 −C until ELISA was performed. The pooled tear specimens were measured using a Mouse VTN (vitronectin) ELISA kit (Fine Test; EM0500), according to the manufacturer’s instructions. Fresh serum samples were isolated by centrifuging the tubes for 15 min at 2000× *g* and were then diluted 1:100 in phosphate-buffered saline (PBS). Serum samples were stored at -20 °C until ELISA was performed.

### 4.11. Statistics

The data were generated from three independent experiments. Comparisons of two groups were made using the Mann–Whitney U test. *p* < 0.05 was considered significant.

## 5. Conclusions

Our studies have revealed that the dysregulated expression ofαv integrins and VTN are relevant to the pathology of macrophage-mediated inflammation during DED progression. Pharmacological inhibition of αv integrins by c(RGDfK) decreases the severity of experimental DED. Further explorations of the expression of RGD-binding integrins in various types of ocular and immune cells may help to develop a more complete understanding of the molecular mechanisms of DED pathogenesis and further support the concept of integrins as important drug targets for DED therapy.

## Figures and Tables

**Figure 1 ijms-22-08410-f001:**
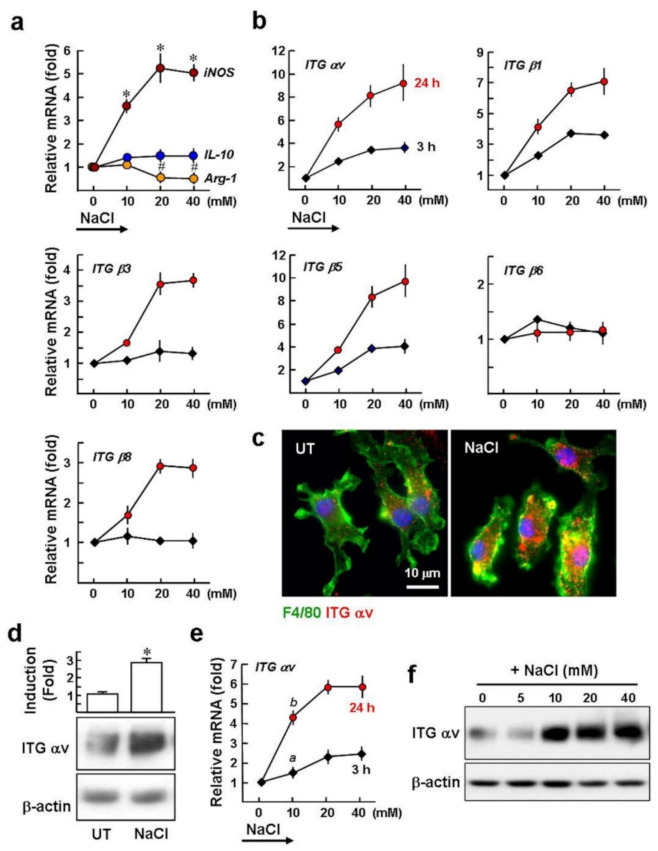
Exposure of macrophages to high salt medium induces expression of iNOS and αv integrins. BMDMs were cultured in isomolar medium (312 mOsm) and switched to hyperosmotic media (343~435 mOsm) by the addition of 10~40 mM NaCl. (**a**) qPCR analysis of the expression of macrophage M1- and M2-associated markers at 24 h. * *p* < 0.0001, ^#^
*p* < 0.02 versus untreated control (cells cultured in isomolar medium). (**b**) qPCR analysis of integrin gene expression at 3 h and 24 h. Data were normalized to *gapdh* expression levels and are summarized as mean ± SE of three separate experiments. After20 mM NaCl treatment for 24 h, the expression of αv integrin was monitored by immunofluorescence staining (**c**) and Western blot analysis (**d**). F4/80: a macrophage marker. Nuclei were stained with Hoechst 33258 (blue). Representative images, Western blots, and densitometric analysis are shown from three independent experiments. * *p* < 0.001 versus untreated BMDMs. (**e**) High salt effect on the αv integrin mRNA expression in THP-1 macrophages. Data were obtained from three independent experiments. *^a^ p* < 0.02, *^b^ p* < 0.006 versus untreated cells. (**f**) Western blot analysis of αv integrin expression in THP-1 after NaCl treatment for 24 h. Representative blots are shown from three independent experiments with similar results.

**Figure 2 ijms-22-08410-f002:**
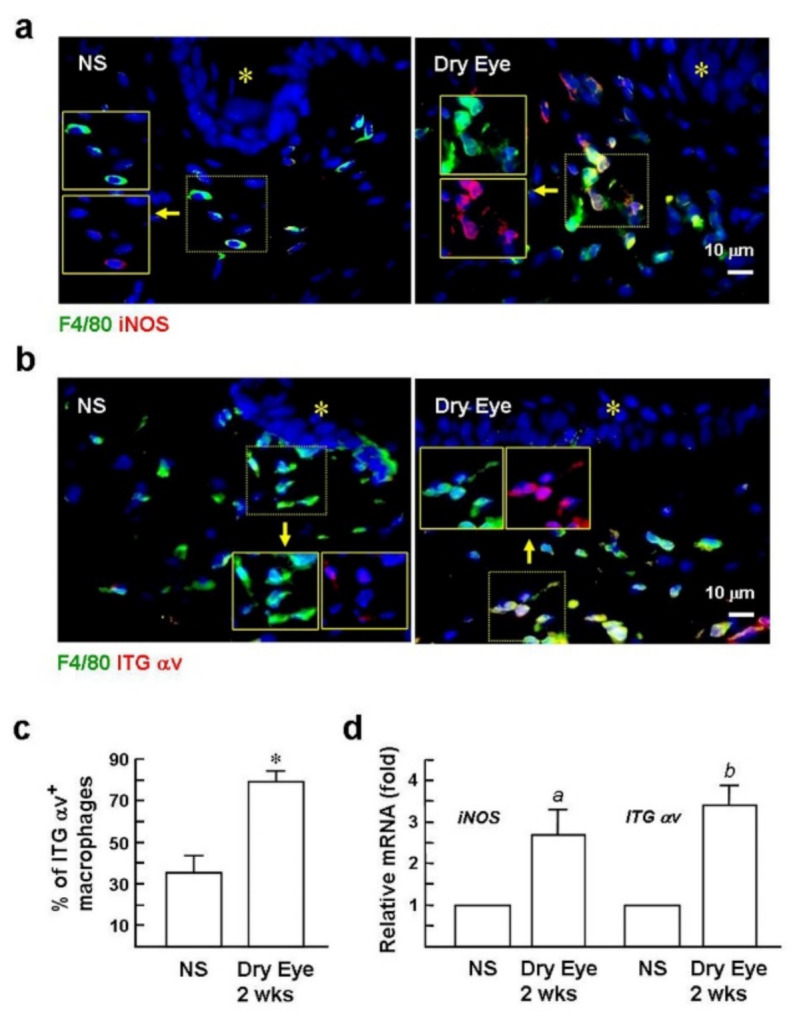
Expression of αv integrin is induced in macrophages in the dry eye. (**a**,**b**) Immunofluorescence staining of iNOS, αv integrin, and F4/80 (green; a macrophage marker) in conjunctiva at day 12 after DED induction. Insets are the images before superimposition. Nuclear counterstaining was conducted using Hoechst 33258 (blue). Asterisks indicate conjunctival epithelium. Representative images are from six sections per mouse eye with six mice per group. NS group: mice housed in a stress-free environment. (**c**) Digital image analysis of the percentages ofαv integrin^+^ macrophages per total macrophages was performed blindly on an average of six randomly selected × 1000 magnification fields from each section (*n* = 6) using a Zeiss epifluorescence microscope and Zeiss software. * *p* < 0.0001 versus NS group. (**d**) qPCR evaluates the levels of iNOS and αv integrin in the conjunctiva after DED induction for 12 days. Values are expressed as mean ± SE. *^a^ p* < 0.03 versus NS mice. *^b^ p* < 0.002 versus NS mice.

**Figure 3 ijms-22-08410-f003:**
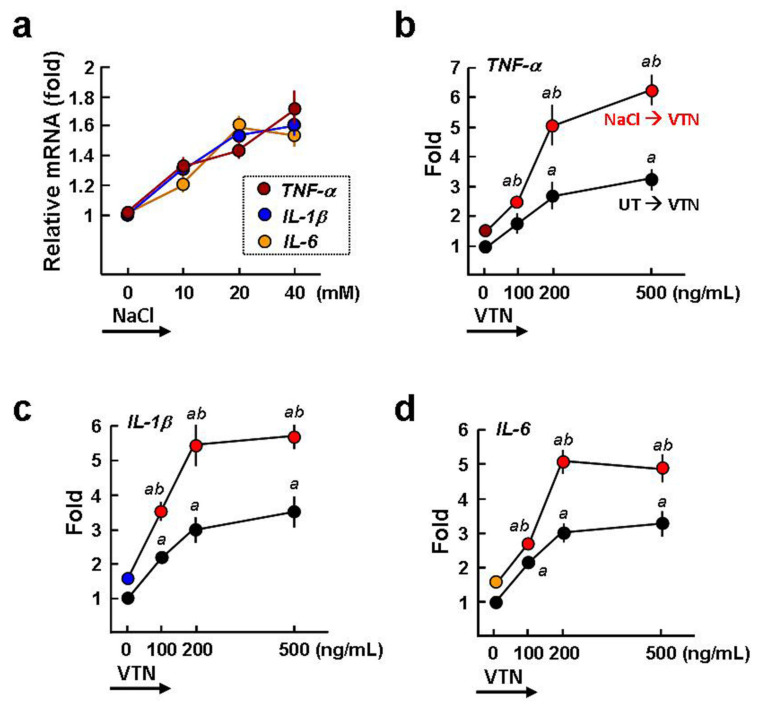
VTN is a macrophage proinflammatory stimulator. (**a**) qPCR analysis of the proinflammatory cytokine expression induced by NaCl for 24 h. Data were normalized to *gapdh* expression levels and summarized as the mean ± SE of three separate experiments. (**b**–**d**) VTN promotes the expression of the *TNF-α*, *IL-1β*, and*IL-6*genesin NaCl-treated BMDMs. Cells were cultured in isomolar medium or hyperosmotic medium by adding 20 mM NaCl for 24 h and were then treated with VTN for a further 3 h before qPCR analysis. Data are from three independent experiments. *^a^ p* < 0.05 versus VTN untreated cells. *^b^ p* < 0.01 versus corresponding UT/VTN-treated cells.

**Figure 4 ijms-22-08410-f004:**
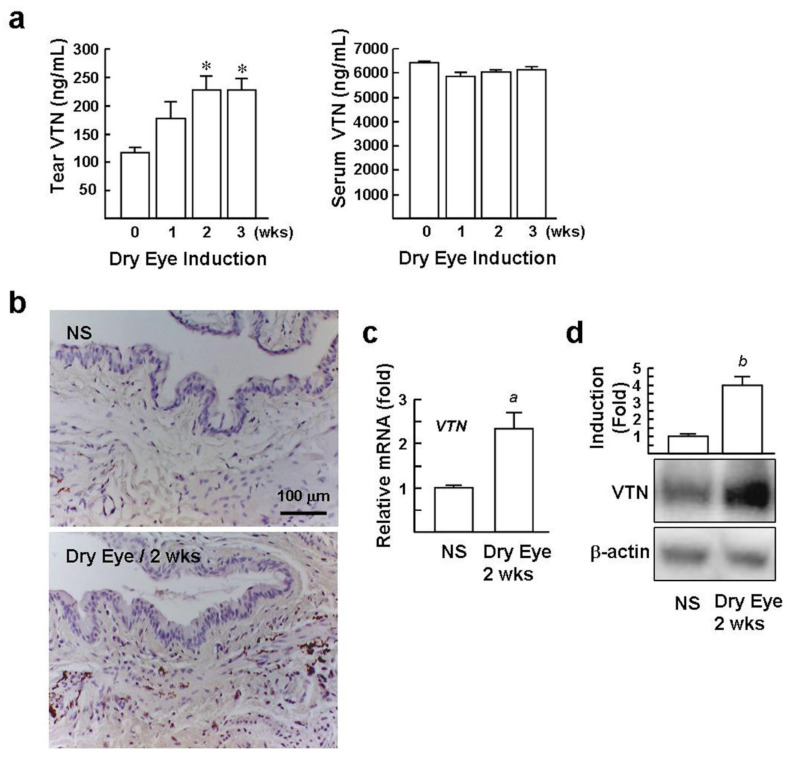
Levels of VTN in the tears and conjunctiva are elevated in DED mice. (**a**) Tear fluid and serum concentrations of VTN in DED mice were assessed using ELISA. Data (mean ± SE) are obtained from 10 mice from day 0 through 3 weeks. * *p* < 0.001 versus day 0. (**b**) Immunohistochemical staining of VTN (brown color) in the conjunctiva. Representative micrographs from two independent experiments (*n* = 6) are shown. Analysis of VTN expression in the conjunctiva of DED and NS mice (each *n* = 6) by qPCR (**c**) and Western blotting (**d**). Representative blots and densitometric analyses of 6 samples are shown. *^a^ p* < 0.0001 versus NS mice. *^b^ p* < 0.0002 versus NS mice.

**Figure 5 ijms-22-08410-f005:**
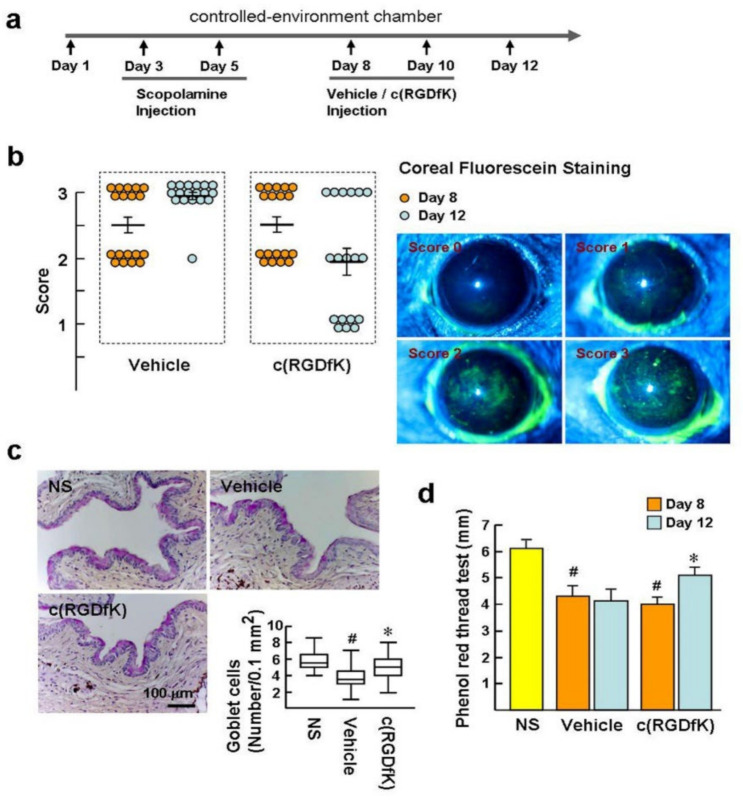
c(RGDfK) ameliorates clinical signs of dry eyes. (**a**) Schedule: To induce DED, C57BL/6 mice were housed at a low-humidity CEC with subcutaneous injection of scopolamine at days 3 and 5. Subsequently, mice received 100 µL c(RGDfK) at a concentration of 100 µM or vehicle (1% DMSO in BSS) through subconjunctival injection at days 8 and 10. Non-stressed mice (NS; *n* = 10) housed under normal ambient conditions served as a control. (**b**) Mean corneal staining score of the mice before (day 8) and after treatment (day 12). Representative images of corneal fluorescein staining were used to estimate corneal damage. (**c**) PAS staining of the goblet cells in the conjunctival fornices at day 12. The representative photographs of goblet cells (pink) and the average number of goblet cells in each group. Data show median/interquartile ranges of individual mice (*n* = 8 per group). (**d**) Analysis of the tear volume using the phenol red thread test expressed in millimeters of thread at day 12. Data are presented as mean ± SE. *n* = 8 per group. ^#^
*p* < 0.001 versus NS group. * *p* < 0.01 versus c(RGDfK) group at day 8.

**Figure 6 ijms-22-08410-f006:**
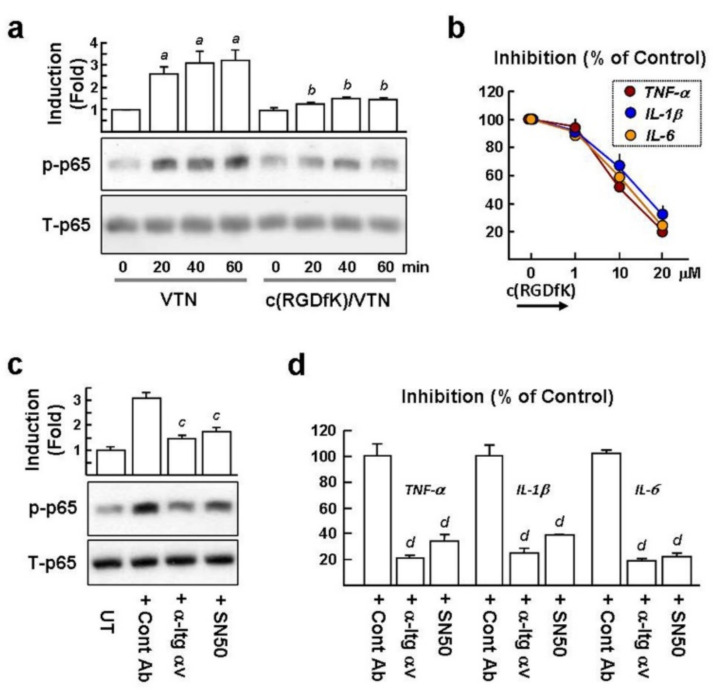
VTN induces NF-κB activation in NaCl-treated BMDMs to upregulate proinflammatory cytokine gene expression. BMDMs were treated with 20 mM NaCl for 24 h and were then treated with 20 µM c(RGDfK), 10 µM SN50, 5 µg/mL of anti-Itg αv antibody, or IgG control for 10 min prior to stimulation with 200 ng/mL of VTN for a further 3 h. (**a**) Western blotting of NF-κB p65 phosphorylation (p-p65) and total p65 stimulated by VTN for 20~60 min. Representative blots and densitometric analyses from three samples are shown. *^a^ p* < 0.001versusuntreated cells. *^b^ p* < 0.005 versus corresponding VTN-treated cells. (**b**) qPCR analysis of proinflammatory gene expression stimulated by VTN for 3 h. Data were summarized as mean ± SE of three separated experiments. (**c**) Representative Western blots of p-p65 and total p65 stimulated by VTN for 20 min and densitometric analyses from three samples are shown. *^c^ p* < 0.002 versus Cont Ab-treated cells. (**d**) qPCR analysis of proinflammatory gene expression stimulated by VTN for 3 h. Data were summarized as the mean ± SE of three separated experiments. *^d^ p* < 0.0001 versus Cont Ab-treated cells.

**Figure 7 ijms-22-08410-f007:**
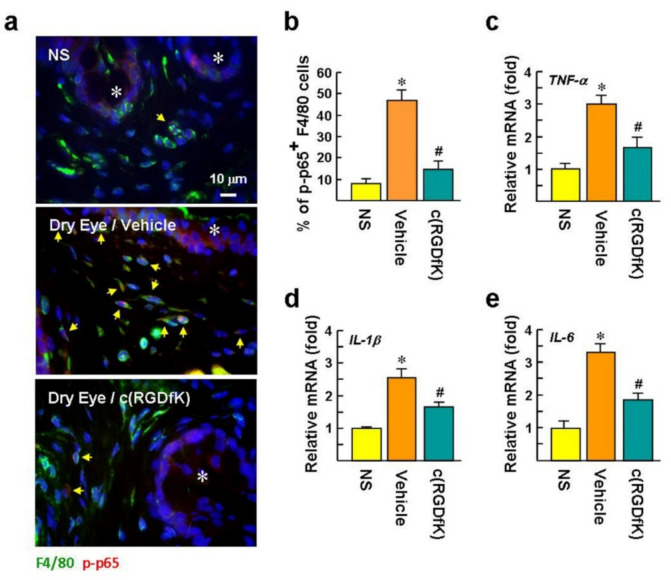
c(RGDfK) suppresses inflammatory responses in dry eyes. The mice received DED induction and c(RGDfK) treatment as described in the legend of Figure 5A. (**a**,**b**) Representative images of immunofluorescence staining of NF-κB p-p65 (Ser536) and F4/80 in the conjunctiva after DED induction for 12 days. The p-p65 located in the cell nucleus is confirmed by Hoechst 33258 counterstaining. Asterisks indicate conjunctival epithelium. Arrows indicate p-p65 and F4/80 double-positive cells. There were 30 randomly selected fields in each treatment (*n* = 6 per group) that were photographed, and the percentage of p-p65 and F4/80 double-positive cells per total F4/80-positive cells was calculated. * *p* < 0.0006 versus NS mice. ^#^
*p* < 0.0001 versus vehicle group. (**c**–**e**) qPCR evaluation of the mRNA levels of proinflammatory cytokines in the conjunctiva after DED induction for 12 days. Values are expressed as mean ± SE. * *p* < 0.001 versus NS mice. ^#^
*p* < 0.02 versus the vehicle-treated group.

**Table 1 ijms-22-08410-t001:** Primers used in the real-time qPCR.

Target Gene	Primer (Sense)	Primer (Antisense)	Accession No.
moItgAV	5′-TGTTCACACTTTGGGCTGTG	5′-GCCTCTATCCAGTCGACCAA	NM_008402.3
moItgB1	5′-GGTCAGCAACGCATATCTGG	5′-AATCAGCGATCCACAAACCG	NM_010578.2
moItgB3	5′-GCTCATTGGCCTTGCTACTC	5′-GTTGTTTGCTGTGTCCCACT	NM_016780.2
moItgB5	5′-GACCTTTCTGCGAGTGTGAC	5′-CTCTCCATGGCCTGAGCATA	NM_001145884.1
moItgB6	5′-GCATTTGGAAGCTGCTGGTA	5′-ATCTGAGGAAAGGCCTGCTT	NM_001159564.1
moItgB8	5′-CTGGGCCAAAGTGAACACAA	5′-CAACTGGACAGCCTTTGCTT	NM_177290.3
moVTN	5′-CATACTAGCCCTGGTGGCAT	5′-CCATGAAACCCTGAGTGCAG	NM_011707.2
moNOS2	5′-CCTTGTTCAGCTACGCCTTC	5′-CTTCAGAGTCTGCCCATTGC	NM_010927.4
moIL-10	5′-AGCTGAAGACCCTCAGGATG	5′-CACTCTTCACCTGCTCCACT	NM_010548.2
moArg1	5′-AAGACAGGGCTCCTTTCAGG	5′-AGCAAGCCAAGGTTAAAGCC	NM_007482.3
moTNF-α	5′-CCAAATGGCCTCCCTCTCAT	5′-CACTTGGTGGTTTGCTACGA	NM_013693.3
moIL-1β	5′-GGCTCATCTGGGATCCTCTC	5′-GTTTGGAAGCAGCCCTTCAT	NM_008361.4
moIL-6	5′-GGAGCCCACCAAGAACGATA	5′-ACCAGCATCAGTCCCAAGAA	NM_031168.2
mGAPDH	5′-AACGGATTTGGCCGTATTGG	5′-CATTCTCGGCCTTGACTGTG	NM_001289726.1

## Data Availability

The data presented in this study are available on request from the corresponding author.

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
