# Peer review of "Integrin αv and Vitronectin Prime Macrophage-Related Inflammation and Contribute the Development of Dry Eye Disease"

_ijms, 2021, doi:10.3390/ijms22168410_

Round 1

Reviewer 1 Report

The possible role of macrophage should be added in Introduction.

Author Response

Reviewer 1: The possible role of macrophage should be added in Introduction.

Ans. We have revised our Introduction to describe the possible role of macrophages.

The sentence in Introduction (lines 45-46) was removed: “Notably, subconjunctival injection of clodronate, to locally deplete the resident conjunctival macrophages, significantly reduces ocular surface damage and proinflammatory cytokine expressionin dry eye.”

The revised description (lines 47-56) is: “Mouse model of desiccating stress-induced dry eye is shown to recruit monocytes into conjunctiva and induce them into activated macrophages [7,8]. The activated macrophages have upregulated expression of genes associated with antigen presentation, cytokine/chemokine, M1 macrophage and NLRP3 inflammasome pathway [8]. These activities further enhance the inflammatory status of ocular surface. The crucial involvement of macrophages in ocular surface inflammation is based on depletion of macrophages. Subconjunctival injection of clodronate, to locally deplete the resident conjunctival macrophages, significantly reduces the generation of autoreactive CD4+ T cells, ocular surface damage and proinflammatory cytokine expression in dry eye [9,10].”

In addition, two references have been cited.

Pflugfelder, S.C.; Bian, F.; Gumus, K.; Farley, W.; Stern, M.E.; De Paiva, C.S. Severity of Sjögren's Syndrome Keratoconjunctivitis Sicca Increases with Increased Percentage of Conjunctival Antigen-Presenting Cells. Int. J. Mol. Sci. 2018, 19, 2760.

Alam, J.; de Paiva, C.S.; Pflugfelder, S.C. Desiccation Induced Conjunctival Monocyte Recruitment and Activation - Implications for Keratoconjunctivitis. Front. Immunol. 2021, 12, 701415.

Reviewer 2 Report

In the manuscript entitled “Integrin αv and Vitronectin Prime Macrophage-Related Inflammation and Contribute the Development of Dry Eye Disease” the authors compile and review the publication landscape of vitronectin (VNT), including the human Vn to generate a translational research of their results for of Dry Eye Disease.

Author Response

We are very grateful for the comments of the reviewer 2

Reviewer 3 Report

Macrophages play an important role in Dry-Eye Disease (DED). Further, it is well known that integrins play an important role in many ocular diseases. Recently, Perez et al. 2016 suggested that integrins in T cells play a role in role in T cells in DED Therefore, it is important to understand the expression of integrins in macrophages under dry-eye conditions.

In summary, the authors demonstrated that:

  1. Macrophages express integrins under hyperosmolarity conditions, specifically ITG alpha-v
  2. Mice macrophages of the conjunctiva upregulate the expression of ITG alpha-v and iNOS
  3. Vitronectin and hyperosmolarity together induced higher inflammation (TNFa, IL-1b, IL-6), as compared to themselves alone
  4. Induction of DED induces vitronectin expression in the tear fluid, serum, and conjunctiva
  5. C(RGDfK) prevented the DED phenotype in mice
  6. Vitronectin increased the expression of NFKB, a potent stimulator of inflammation, and C(RGDfK) blocked this effect (both in BMDMs and dry-eye conditions)

Notably, expression of NFKB by vitronectin and blockage of NFKB via C(RGDfK) are very important findings. Overall, these important findings suggest that integrins maybe an important therapeutic target for DED.

Minor suggestion

On line 87-88, the authors state that, “ In addition, 40 mM NaCl treatment did not induce cell death, 87 as monitored by a trypan blue exclusion test (our unpublished data).” – Please remove our unpublish data and replace with “data not shown”.

Author Response

Reviewer 3: Macrophages play an important role in Dry-Eye Disease (DED). Further, it is well known that integrins play an important role in many ocular diseases. Recently, Perez et al. 2016 suggested that integrins in T cells play a role in T cells in DED. Therefore, it is important to understand the expression of integrins in macrophages under dry eye conditions.

Ans. We agree the reviewer comment for the report by Perez et al. The report indicates that a T cell surface receptor, lymphocyte function-associated antigen 1 (LFA-1; integrin αLβ2), plays crucial roles in T cell activation to promote the development of DED. The report has been cited and discussed in our revised manuscript.

The description in Discussion (line 346-350) is: “In addition, drug discovery research in the anti-integrin therapy includes the lymphocyte function-associated antigen 1 (LFA-1; integrin αLβ2; a T cell surface receptor). In this regard, Lifitegrast blocks the association of LFA-1 with intercellular adhesion molecule 1 (ICAM-1), leading to alleviate T cell-mediated inflammatory responses in DED [31].”

Minor suggestion: On line 87-88, the authors state that, “In addition, 40 mM NaCl treatment did not induce cell death, 87 as monitored by a trypan blue exclusion test (our unpublished data).” – Please remove our unpublished data and replace with “data not shown”.

Ans. The sentence has been corrected.